# The Strange Case of Dr. Moloch and Mr. Snazzo (or the Parmenides' Riddle Once Again)

Alberto Voltolini

Department of Philosophy and Education Sciences, University of Turin, 10124 Turin, Italy; alberto.voltolini@unito.it

**Abstract:** Once one draws a distinction between loyal non-existent items, which do not exist in a non-universal sense of the first-order existence predicate, and non-items, which fail to exist in a universal sense of that predicate, one may allow for the former but not for the latter in the overall ontological domain, so as to adopt a form of soft Parmenideanism. There are both theoretical and empirical reasons for this distinction.

**Keywords:** fictional entities; concrete entities; empty names; existence





## 1. Introduction

In this paper, I first want to show that it is theoretically recommendable to endorse an ontologically syncretistic stance as regards the Parmenidean riddle of non-existence; namely, a stance that lies in between ontological luxuriantists *à la* Meinong and ontological anti-luxuriantists *à la* Parmenides. For syncretists, on the one hand, items that fail to exist in a first-order and non-universal sense of «existence», or, which is the same, that are not captured by a restricted application of the first-order yet universal existence predicate, can be admitted in the overall ontological domain. Following [1]'s usage, let me call them *loyal non-existent items*. For syncretists, on the other hand, *non-items* that fail to exist in a first-order yet universal sense of «existence», or, which is the same, that are not captured by an unrestricted application of that predicate, can utterly be ruled out of that domain. Properly speaking, with respect to them, utterly empty genuine singular terms are the only things that there are. So, my syncretistic account can be taken as a form of soft Parmenideanism in the spirit (but not in the letter) of [2]'s account.

Second, this syncretistic stance seems to be corroborated by some empirical evidence. For ordinary people draw different assessments when they are asked to compare, as regards existence, on the one hand, loyal non-existent items with (actual) concrete beings, and on the other hand, again loyal non-existent items yet with non-items.

More specifically, Section 2 theoretically argues in favor of ontologically endorsing loyal non-existent items while rejecting non-items. Section 3 gives empirical support in favor of this ontological distinction. Section 4 concludes by dismissing all other extant interpretations of the empirical findings, either anti-luxuriantist or luxuriantist, as less plausible than the syncretistic interpretation.

## 2. Loyal Non-Existent Items vs. Non-Items

In his poem *On Nature*, Parmenides notoriously exposes his riddle on non-existence in the following terms:

"Come now, I shall tell—and convey home the tale once you have heard—/just which ways of inquiry alone there are for understanding:/the one, *that [it] is and that [it] is not not to be*/, is the path of conviction, for it attends upon true reality,

/but the other, *that [it] is not and that [it] must not be/*, this, I tell you, is a path wholly without report: /for neither could you apprehend what is not, for it is not to be accomplished/, nor could you indicate it." ([3]: Fr.2; my italics).

Famously, in his [4] (p. 83) (see also [1] (p. 224)), Meinong has tried to solve the riddle by providing an argument against the Parmenidean position that can be reconstructed as follows [5] (p. 6). In it, premise 1) directly attacks Parmenides' final claims in the above quotation, in order, *contra* Parmenides, for non-beings to be ontologically allowed:

(1)   In order to merely *say* truly that a non-existent object does not exist, one must designate *it*, by expressing a sentence about *it* (*mutatis mutandis*, the same holds of thinking truly that a non-existent object does not exist);

(2)   In order to so designate that object, that object must already be admitted within the overall ontological domain;

(3)   Hence, in order to merely say truly that such an object does not exist, such an object must already be admitted within the overall ontological domain.

To me, Meinong's argument can be accepted *only partially*. Indeed, on the one hand, it is convincing insofar as we either intuitively accept, or better, we ontologically argue for their acceptance, *loyal non-existent items*, as I will label them following Meinong himself (1971). Loyal non-existent items make premise (1) in Meinong's argument true since they do not exist only in a first-order and *non-universal* sense of the existence predicate; typically, *having causal powers* or *having spatiotemporal determinations* [5–8], which only (actual) concrete beings satisfy. Or, which is the same, loyal non-existent items do not exist only if the extension of the first-order yet different *universal* existence predicate, the one meaning *being an object*, or *being identical with something* [5,7,8] is restricted to a subdomain of the overall ontological domain to the items that exist in the above different non-universal sense: as I just said, (actual) concrete beings[1] "Non-universal" in this sense means that the predicate applies only to *some* items in the overall ontological domain, the totality of all beings, or, to speak in non-nominalist terms, it expresses a property that is instantiated by such items only; "universal" means the opposite. In this respect, *pace* Salmon [10] and van Inwagen [11], there is nothing really paradoxical or unintelligible in the famous Meinong's (1960:83) motto, «there are things of which it is true that there are no such things», *when meant as saying* that *there are things that do not exist* (in the above non-universal sense). Paradigmatic examples of loyal non-existent items are future objects, merely possible objects, and even fictional objects[2] For they do not exist in the above sense since they so exist either merely in the future, or merely possibly, or even merely non-spatiotemporally [14–16].

Yet, on the other hand, the argument is not convincing if one wants to extend Meinong's reply as covering 'things' that do not exist in the other, still first-order yet universal, sense of the existence predicate, or, which is the same, as 'things' not falling in the unrestricted extension of that existence predicate; *non-items*, as one may label 'them' (I use this terminology in order to distinguish these 'things' from what Routley's [17] unfortunately labeled *non-entia*, which are meant to include both loyal non-existent items and non-items). For, if one applies Meinong's argument to non-items, one obtains the genuine paradox Parmenides wanted to dispense with; namely, that non-items that do not belong to the overall ontological domain actually belong to it.

(The) Nothing is the paradigmatic example of this paradox. As some [18–20] claim in a Meinongian vein, in order to truly say or think that (the) Nothing does not exist, in the first-order yet universal sense that it is not an object or it is not identical with something, one must designate it, hence postulate that it is an object as well.

However, since the existence predicate that is involved in the relevant existential sentence allegedly about (the) Nothing:

(1) (The) Nothing does not exist

Is supposed by the Priest-like Meinongian to mean the above first-order yet universal property of existence or to occur utterly unrestrictedly, why should we be attracted by such a siren song and get contradictory things such as (the) Nothing in one's ontology; namely, things that are such that they both are and fail to be objects? Twardowski, for one, thought that such contradictory things must be dispensed with. For not only are they implausibly contradictory, but their contradictoriness prevents the overall ontological domain from being constituted once and for all. Indeed, says [21] (pp. 19–20), in order for the complement of a set to be successfully conceived, one must have a superordinate concept under which both the members of that set and the members of that set's complement fall, e.g., Greeks (i.e., individuals falling under the concept of *being Greek*) and non-Greeks (i.e., individuals falling under the opposite concept of *being non-Greek*) are both human beings (i.e., individuals falling under the superordinate concept of *being a human*). Yet, one cannot have a superordinate concept under which both the somethings, i.e., the inhabitants of the overall ontological domain, and the non-somethings fall. For that concept would be both superordinate to the somethings and coordinated with it, something again[3].

If one wants to avoid the above paradox, then one is forced to conclude that there are no non-items at all. Properly speaking, in order to avoid the paradox from surfacing again in saying that non-items do not exist, ontologically rejecting non-items properly means that the only things that there really are with respect to 'them' are the utterly empty linguistic terms that are mobilized in the erroneous conviction that they designate 'them'; in the above case, the term "(the) Nothing". Following a standard tradition (e.g., [23], one may label *genuine singular terms* as terms that purportedly designate items. Given this qualification, one may take *utterly empty genuine singular terms* as terms that do not designate anything at all. Now, independently of the case of "(the) Nothing", there are other cases of terms that fall under this subcategory of *utterly empty genuine singular terms*, which fail to designate non-items, even of a more modest kind than (the) Nothing. First of all, the *misunderstanding case*. Consider the case of «Max», as a case of an expression misheard as a name, in misunderstanding someone else's utterance of «Go to the max!»; so misheard, "Max" does not refer to anything [24] (p. 156). [25] (p. 70)'s «Moloch»- case is another case in point; a word that is actually a noun (synonymous with «king»), yet it is misunderstood as if it were a name of a deity. Granted, one may take these cases as merely seemingly involving genuine singular terms since one misunderstands as genuine singular terms linguistic terms that actually belong to other linguistic categories. Yet, moreover, the *made-up case* is a case that does not raise the above problem, for the terms it involves are utterly empty genuine singular terms in the above sense. Consider [25] (p. 81)'s «Snazzo», a name invented by Kripke himself to allegedly refer to something and yet that nobody has used, not even in the attempt at meaning a *fictum* (or even, as Kripke also says, as meaning a *fictum fictum*, a fictional character that is generated by a story in a story). This case, and possibly also the previous one, is completely different from the «Newman1»- [26], the «Elip»- [27], and the «Holmes» cases [12,25,28–34][4]. For in these latter cases, depending on one's ontological taste, one may take such names as referring to a future entity, to a merely possible entity, or to a fictional entity, respectively, i.e., to non-existent loyal items. However, in the former case, one only has an utterly empty name at one's disposal[5].

Thus, as regards non-items, one may well reject premise (1) of Meinong's argument in one of the classic ways in which it has been disputed. First, one may claim *à la* [36] that once genuine singular terms purportedly for non-items are analyzed in terms of Russell's theory of descriptions, then against premise (1), one may truly say that such an item does not exist without committing oneself to one's designating it [5] (p. 24). This account straightforwardly applies to "(the) Nothing" [37]. One may paraphrase (1) above in Russellean terms as follows:

(1R) There is no thing which is such that it is identical with nothing, i.e., that there is no thing that is identical with it.

Where the seeming appearance that "(the) Nothing" designates something is dissolved since that term no longer figures in the paraphrase, nor can one legitimately say that such a term has a Russellean denotation, i.e., something uniquely satisfying the description that term amounts to.

Or second, one may claim *à la* [38] that, once the relevant true negative existential is analyzed in metalinguistic terms, the only things that there really are in the problematic cases raised by such existentials are utterly empty genuine singular terms. Against premise (1), this solution straightforwardly applies to the above cases of genuinely empty names, e.g.,:

(2) Snazzo does not exist

(2F) "Snazzo" does not refer[6,7].

Now, if one combines the acceptance of Meinong's argument with respect to loyal non-existent items with the rejection of such an argument with respect to non-items, one obtains a form of soft Parmenideanism in [2]'s terms. Unlike non-items, loyal non-existent items are not ruled out of the overall ontological domain. For one can designate them and think of them insofar as they fail to exist merely in the non-universal first-order sense of existence, yet they still exist in the universal first-order sense[8].

### 3. The Distinction's Empirical Support

At this point, one may raise an immediate objection to my ontological distinction between loyal non-existent items and non-items. For ontological anti-luxuriantists reprising Parmenides in the wake of Quine [43], the difference between loyal non-existent items and non-items is unjustified. According to them, the overall ontological domain has an extension that is restricted to items that exist in my alleged non-universal sense: neither merely possible objects nor fictional objects belong to the overall ontological domain, for neither intuitive consideration nor ontological arguments in favor of them are satisfying[9]. Hence, for them, there is no chance for Meinong's argument to allow for at least loyal non-existent items, for there are no such things either. Indeed, for them, the case of genuine singular terms such as «Newman1», «Elip», and «Holmes» must be equated to the case of utterly empty genuine singular such as «Moloch» and «Snazzo».

Yet, if anti-luxuriantists are not moved by ontological arguments in favor of loyal non-existent items, perhaps they will be impressed by empirical findings that suggest that the case of loyal non-existent items is different from that of non-items so as to block merely the ontological allowance of non-items, even of a modest kind such as Moloch and Snazzo. For these findings allegedly show that ordinary people are ontologically committed to loyal non-existent items but not to non-items. These findings concern the truth evaluation that is ordinarily given to comparative existentials of certain sorts [44][10].

On the one hand, ordinary people take the following kind of positive comparative existentials as false:

(3) Emma Bovary exists, just as Angela Merkel and Vladimir Putin

While they take the following kind of positive comparative existentials as true:

(4) Lady Gaga exists, just as Angela Merkel and Vladimir Putin
(5) Jacinda Ardern exists, just as Angela Merkel and Vladimir Putin
(6) Donald Trump exists, just as Angela Merkel and Vladimir Putin.

Yet, on the other hand, they also take the following kind of positive comparative existentials as false:

(7) Emma Bovary exists, just as Moloch and Snazzo

While they take the following kind of positive comparative existentials as true:

(8) Emma Bovary exists, just as Anna Karenina and Emma Woodhouse
(9) Emma Bovary exists, just as Desdemona and Othello
(10) Emma Bovary exists, just as Sherlock Holmes and Tom Sawyer.

In my interpretation of the data, one and the same first-order universal property of *existence* is predicated in the relevant comparisons, but either in a restricted or unrestricted way (which, as we saw in the previous Section, is the same as appealing both to a first-order non-universal and to a first-order yet universal different properties of existence). Hence, on the one hand, with respect to a restricted context, viz. domain of (actual) concrete beings, the one a sentence like (3) mobilizes, (3) is taken to be false. For in that context, existence only holds for such beings since the domain in question is just the restricted domain of (actual) concrete beings. Yet, on the other hand, with respect to the non-restricted context a sentence like (7) mobilizes, the context appealing to the overall ontological domain, (7) is also taken to be false. For existence also holds of fictional objects, insofar as, *qua* loyal non-existent items, they figure in that overall domain, but not of 'anything' outside that domain, i.e., non-items. For properly speaking, that 'anything' is not a thing at all since, as I said before regarding non-items, the only things that there are, properly speaking, are just genuinely empty names like «Moloch» and «Snazzo».

From the point of view of the philosophy of language, this interpretation is a *relativistic* interpretation. For it holds that the relevant existence predicate always means the same thing, i.e., the very same first-order-yet-universal property of *being an object* (or of *being identical with something*), yet the sentences in which that property is predicated *are evaluated with respect to different contexts* in the sense of different domains of items whose extension thereby differs: notably, the restricted domain of (actual) concrete beings and the unrestricted overall ontological domain. Only a first-order, non-universal property of existence and a first-order universal property of existence are required to yield the restricted and the unrestricted domain, respectively [45,46].

This relativistic way of interpreting the data has a larger explanatory power. For it may also be appealed to in other close situations that involve comparisons but do not mobilize existence. Consider first:

(11) Boris Johnson dances, just as Rudolph Nurejev.
(12) Boris Johnson dances, just as that turtle over there.

Both (11) and (12) can be evaluated as false since the extension of the predicate *being a dancer* is restricted to elegant dancers in the first case and unrestricted in the second case. Second, consider a case inspired by Charles Travis' original example of Pia [47]. Suppose that Pia paints in green some of the russet leaves of her plants while painting in another color, say violet, some of her russet pears. If one chromatically compares such leaves with such pears, one obtains a sentence that is likely false:

(13) Pia's leaves are as green as her pears.

Yet, suppose now that Pia's artifactually modified leaves are chromatically compared with some vegetables that grow in her garden without any depictive intervention, say, Pia's naturally violet aubergines. One thus obtains another sentence that is likely false just as (13):

(14) Pia's leaves are as green as her aubergines.

In my interpretation of this case, while (13) tends to be evaluated by folks as false with respect to a context, viz. a restricted domain containing only artifactually painted objects, some of which—the leaves—are artifactually green while some others—the pears—are artifactually violet, (14) tends to be evaluated by folks as false with respect to a context, viz. an unrestricted domain that contains both artifactually painted green objects—the same (in actual fact, naturally russet) leaves—and naturally colored violet objects—the aubergines.

## 4. Objections and Replies

So far, so good. Yet, the fact that the above data can be interpreted as I did does not rule out that other interpretations of them are possible, where such interpretations do not appeal to a distinction between loyal non-existent items and non-items because either they are anti-luxuriantist or luxuriantist interpretations. However, such interpretations are less plausible than the one I provided, which is, for me, the best explanation for the data. Or so I claim.

First of all, on behalf of anti-luxuriantists, one may object that (7) is taken to be false for its first conjunct is already false since Emma does not exist, this also being the reason why (3) is false.

Yet, first, this objection does not explain why folks take (8)–(10) as true; they should take them as false as well, since (8)–(10) share with (7) and (3) their first conjunct. Nor second, does it take into account the fact that the *negative* existential corresponding to (3) would also be taken to be false; for anti-luxuriantists it should count as true since it contains the negations of (3)'s conjuncts:

(3N) Emma Bovary does not exist, just as Moloch and Snazzo.

Moreover, still on behalf of anti-luxuriantists, one may say that «to exist», when predicated on names such as «Emma Bovary», «Anna Karenina», «Desdemona», and «Sherlock Holmes» must be interpreted as «to exist in fiction [meaning 'in some fiction or other']». This is why both (3) and (7) are assessed to be false, for neither Angela Merkel and Vladimir Putin, nor Moloch and Snazzo, exist in fiction.

However, if this were the case, then if we had:

(15) In fiction, Emma Bovary exists, just as Napoleon and Kutuzov

This sentence should be evaluated as true, not as false as one may expect in analogy with (3), for both Napoleon and Kutuzov are also fictional protagonists, as any reader of Tolstoy's *War and Peace* knows.

Furthermore, a different interpretation of the findings coming from the opposite, luxuriantist, side may come to the fore. A philosopher of a particular kind, namely, a Meinongian–Platonic anti-Parmenidean fan of the idea of *ways of being*, might say that the findings show that ordinary people rank items according to their different degrees of existence. More precisely, she would say, folks take that it is false both that Emma Bovary exists like Merkel and Putin and that she exists like Moloch and Snazzo, for on the one hand, she does not have existence to the same degree as Merkel and Putin, since unlike them she is not an (actual) concrete being, yet on the other hand, given her abstract mode of being, she still has a higher degree of existence than Moloch and Snazzo. More generally, according to this luxuriantist interpretation, in the overall hierarchically organized chain of beings, on the one hand (actually, *pace* Plato) (actual) concrete objects have a mode of being higher than that of abstract objects, as fictional objects are metaphysically often taken to be. Yet, on the other hand, *qua* abstract objects, fictional objects have a mode of being still higher than non-items like Moloch and Snazzo.

Actually, from the point of view of the philosophy of language, this interpretation of the findings is a *contextualist* interpretation (e.g., [48]). For it holds that the relevant predicate of existence *changes its meaning*—from meaning *concretely existing* to meaning

*abstractly existing*—in the different comparisons mobilized by the respective sentences, (3) and (7) in our case. As if such sentences, respectively, meant:

(3C) Emma Bovary exists *in the same concrete way* as Angela Merkel and Vladimir Putin.

(7C) Emma Bovary exists *in the same abstract way* as Moloch and Snazzo.

Yet, not only does this contextualist interpretation overintellectualize the reactions of ordinary people, by ascribing to such people a sort of implicit metaphysics of beings that they may well not possess. However, also, it attributes to non-items a sort of existence, the lowest one in the supposed chain of being, where this attribution is not supported by the findings. It is a philosophical conception to hold that once a predication occurs in a sentence, *there is something to which* the property that is so predicated, a property of existence in this case, applies, by simultaneously postulating three modes of being in our case that, respectively, amount to three different species of the *existence* property as a common genus, i.e., *existence as a concrete object*, *existence as an abstract object* and *existence as a non-item*, whatever this means. However, it is hard to ascribe such a philosophical conception to ordinary people.

At this point, a different sort of objection may arise, which attempts to question the coherence of my interpretation. According to my interpretation, precisely because I do not ontologically allow for non-items and I say that, where they are concerned, the only things that really belong to the overall ontological domain are the utterly empty genuine singular terms that supposedly designate such non-items but in fact, fail to designate anything whatsoever, (7) must be suitably reinterpreted as well. For its two conjuncts must, respectively, have an ordinary and a metalinguistic meaning, as if (7) overall falsely said that Emma Bovary exists while *«Moloch» and «Snazzo» refer*. Yet, how can (7) have this meaning, given the hidden anaphora it contains regarding the existence predicate, explicitly occurring in its first conjunct and implicitly occurring in its second conjunct?

However, the use-mention shift occurring in (7) is quite compatible with the hidden anaphora it contains. For it also occurs in completely different sentences, such as, e.g., [49] (p. 412):

(16) As soon as he asked, «Where is Jane?», she arrived.

Yet, if I acknowledge that (7)'s second conjunct must be properly interpreted metalinguistically, the anti-luxuriantist may come back again and say that this metalinguistic interpretation must be appealed to across the board. More precisely, for her, (3) and (7) would, respectively, mean:

(3R) «Emma Bovary» refers, just as «Angela Merkel» and «Vladimir Putin».
(7R) «Emma Bovary» refers, just as «Moloch» and «Snazzo».

Unlike (3R), taken to be false allegedly because of the falsity of its first conjunct, (7R) would be assessed to be false because of the falsity of both conjuncts, the anti-luxuriant may say.

Yet, if this were the case, then once metalinguistically reinterpreted the three aforementioned sentences (8)–(10) should be evaluated as false for the same reason (falsity of both conjuncts), not as true. Yet, as we saw, this is not the case.

However, this reply only leads the anti-luxuriantist to a more sophisticated variant of her last metalinguistic interpretation of the findings[11]. In order to metalinguistically account for the falsity of (3) and (7) due to the mere falsity of just one of their conjuncts, an anti-luxuriantist may take them as:

(3R′) «Emma Bovary» refers in the same way as «Angela Merkel» and «Vladimir Putin».
(7R′) «Emma Bovary» refers in the same way as «Moloch» and «Snazzo».

Meaning that while «Angela Merkel» and «Vladimir Putin» *fully* refer, «Emma Bovary» *only purportedly*, or even *makebelievedly*, refers to something, while «Moloch» and «Snazzo» fail to refer at all, thus replacing ontological differences by means of kinds of referential differences that take «Emma Bovary» as different wrt one referential mode from «Angela Merkel» and «Vladimir Putin» and with respect to another referential mode from «Moloch» and «Snazzo».

Yet, if it is hard to ascribe folks a commitment to modes of being, it is even harder to ascribe them a commitment to modes of referring! The idea of different modes of referring, as possibly ordered in terms of different degrees of referential success, is not something that can easily be attributed to ordinary people, for whom reference is at most an 'all or nothing' affair[12].

The scrutiny of this last interpretation concludes my survey of the alternative interpretations of the findings. Now, I do not want to rule out that other possible interpretations of such findings may arise. However, at least, among all the extant interpretations, the soft Parmenidean one looks to be the most convincing.

## 5. Conclusions

As often happens in philosophy, in his ontological anti-luxuriantism Parmenides was partially right and partially wrong. Against luxuriantists *à la* Meinong, he was right in ontologically disallowing non-items, for assuming them entails endorsing problematic contradictory claims. However, he was wrong in also ontologically disallowing more mundane loyal non-existent items, for assuming them does not entail any contradiction. Moreover, since the ontologically syncretistic position stemming from both ontologically allowing loyal non-existent items and rejecting non-items is the most intuitive one, as some empirical findings suggest, the burden of the proof is on her to show that this more sensible position is incorrect[13].

**Funding:** This research received no external funding.

**Conflicts of Interest:** The author declares no conflict of interest.

## Notes

[1]    I speak of *actual* concrete beings for, according to [7,9], also merely possible entities are concrete beings, if being concrete is defined as possibly having causal powers or spatiotemporal determinations.

[2]    Insofar as one metaphysically takes fictional objects as instances of abstract objects (see, e.g., [12,13]), one may take that abstract objects in general are cases of loyal non-existent objects. For all of them do not exist spatiotemporally.

[3]    Some may object that *being an intentional object* could be the desired superordinate concept, since even (the) Nothing is something that one can think of (I owe this objection to an external referee). Yet, as [22] has convincingly shown, *being an intentional object* does not single out a metaphysical category under which things may fall, since an intentional object is a schematic object, i.e., something that has no metaphysical nature insofar as it is thought of.

[4]    Granted, one may say that Snazzo nowadays is a loyal non-existent item as well, since it has appeared in [25], as if Kripke had created it as a new abstract artefact [35] (p. 319fn.19). However, this is just a problem with the example, which shows not an impossible ontological 'jump' from non-items to items, but simply that what originally was an utterly empty name now is a name referring to a loyal non-existent item. In order to avoid any confusion, [35] discusses the sentence «Rudyard does not exist» containing his made-up name «Rudyard».

[5]    Leaving the case of (the) Nothing aside, some people (e.g., [5,7]) take standard impossible objects to be loyal non-existent objects. Yet, since it is highly controversial that there are *impossibilia*, it is safer to take the genuine singular terms allegedly referring to them as utterly empty genuine singular terms. In this respect, it would be improper to say that non-items are impossible objects. For 'things' like Moloch and Snazzo are non-items not in the sense that that they are *impossibilia*, but in the sense that the terms allegedly for them are utterly empty genuine singular terms.

[6]    This solution is certainly better than [39]'s logical solution to problem of utterly empty genuine singular terms, which assigns them the very same arbitrary referent, so that «Moloch» turns out to be co-referential with «Snazzo», making an identity sentence such as «Moloch = Snazzo» paradoxically true.

7    Another option is to stick to Negative Free Logic in order to have sentences like (2) true, although "(the) Nothing" is an utterly empty genuine singular term (cf. e.g., [40]). Nowadays, another account has become available that is compatible with the two accounts presented in the text, a phenomenological-fictionalist account. According to it, one may claim *à la* [14] and [41] that the relevant negative existential is false only within the scope of a phenomenological pretense that there is the object the relevant genuine singular term designates within that scope [42].

8    My own version of soft Parmenideanism, however, differs from the one Mumford defends. For according to him, loyal non-existent items must ontologically be ruled out just as non-items (not accidentally, Mumford calls all of them "non-beings"). For he does not allow for the distinction I am advocating here between a first-order universal and a first-order non-universal sense of existence. Thus for him, thinking that a loyal non-existent item does not exist is contradictory [2] (pp. 53,135). Granted, by following [30], he appeals to a distinction between aboutness and reference and claims that loyal non-existent items are objects of aboutne8s but not of reference (ib:146). Yet, for [22,23], aboutness is merely a phenomenological notion that has neither a metaphysical or an ontological counterpart.

9    In actual fact, Quine allowed for some instances of abstract objects (namely, sets), so in a sense he is not an anti-luxuriantist. For he does not restrict the overall ontological domain merely to items that spatiotemporally exist. However, let me put this complication aside.

10   The experiment went as follows. 107 participants took part to the study [MA = 23.76 years; SD = 6.18; 94 female]. Participants were all native Italian speakers. Forty-five written sentences were generated in Italian (randomly presented). Each sentence consisted in a direct existence comparison of the form "X exists as Y exists". X was always a proper name purportedly referring to a fictional character. 5 items included literary fictional characters—e.g., "Dr. Frankenstein", "Ron Weasley"—5 items included fictional characters taken from comic books—e.g., "Mickey Mouse", "Spiderman"—while 5 fictional characters were from classic tales—e.g., "Rapunzel", "Snow White". While the 15 fictional characters in X were kept constant across items, the proper name in Y was manipulated in order to generate three experimental conditions: in 15 of the cases Y included a proper name purportedly referring to a *fictum*—e.g., "Alladin", "Merlin"—15 proper names referred to (actual) concrete individuals—*realia*, just to give them a single name—e.g., "Elon Musk", "Barack Obama"—while 15 proper names were utterly non referring; namely, names such as "Abladin" or "Cerlin" failing to refer to anything whatsoever. In this way, direct existential comparisons were generated in three conditions: 15 *ficta* vs. *ficta* (FF), 15 *ficta* vs. *realia* (FR) and 15 *ficta* vs. non-items, where by talking of "non-items" one wants to stress that the corresponding names do not refer at all (FnI).

11   I owe this interpretation to Andrea Bianchi.

12   Granted, one may take [50]'s notion of a *block* in the origin of the referential chain articulating the use of an utterly empty name as a plausible theoretical explanation of the folks' attestation that such a name does not refer *tout court*. However, if one claimed that there are different modes of blocking, this claim would again be a form of hyperintellectualization of the folks' referential behavior.

13   This paper has been originally presented at the workshops *5th Parma Workshop on Semantics and Pragmatics*, Dipartimento di Discipline Umanistiche, Sociali, e delle Imprese Culturali, University of Parma, 8 October 2021, Parma; *Workshop on Nonexistent Objects*, University of Bochum, 3–4 June 2022, Duisburg; *Metaphysics and Semantics of Fiction*, University of Santa Catalina, 15–17 June 2022, https://us02web.zoom.us/j/89231639292, accessed on 1 May 2023; *Ita-Ont 6*, Dipartimento di Scienze Umane, University of L'Aquila, 20–23 June 2022, L'Aquila. I thank all the participants for their important questions: I also thank Carola Barbero and Fred Kroon for their insinghful comments.

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
