# Peer review of "The Strange Case of Dr. Moloch and Mr. Snazzo (or the Parmenides’ Riddle Once Again)"

_philosophies, doi:10.3390/philosophies8040054_

Round 1

Reviewer 1 Report

I enjoyed the paper very much, and I give some comments coming from a committed anti-luxuriantist! Hope this is helpful.

As far as I know, the proposal is original in that it is a tripartition of the ontological space. Though, it reminded me of the impossible-type of meinongian (like Priest-Berto) because they also have a tripartition: individuals from the actual world, from non-actual possible worlds, from impossible worlds. And given that it is commonly acknowledged that all impossible worlds are further away from the actual world than any possible world, you have a tripartition. Of course, the motivations are not yours. And maybe these people could interpret your empirical results treating "exist" as "exist in some possible world" -- and "not exist" as "exist in some impossible world". You might want to distance yourself from this kind of re-interpretation.
The alternative picture is of course a bipartition of the ontological space, with the existing and nonexisting things. But of course, it does not prevent one from distinguishing several kinds in the non-existing things (as within the existing things, for that matter). And the obvious way to do it is to appeal to the origin of the nonexistent items: they can be fictional, merely possible, dream-based, failed posits, figment of the imagination, misspellings, etc. I guess what you want to make out of these empirical results is that the existence/nonexistence distinction cuts through this domain of "nonexistent objects", and so that we should not be ontologically prejudiced.
Put in such terms (tri or bi-partition), the disagreements tend to become verbal of course. I am not sure the empirical results can really help here. Because someone disagreeing with you will be tempted to re-interpret "exit" in some technical jargon. The only thing that is uncontroversial about the empirical result is that there is a difference between existing/actual objects and the rest; and there are differences between fictional objects and nonfictional objects. But I do not see the strong pull of natural language comparative judgements yet.

As for the more technical points about the quantification domain (you against the luxuriantist and anti-luxuriantist). To my mind, what you use against Meinong is really a reductio (more than a paradox) and I tend to think it is still threatening your position: starting with a domain of existents, Meinong says look, we can say "X does not exist", so X should designate something. You end up with 2 domains (and 2 pairs of quantifiers). But then, look, we can say "X does not subsist" (or any cooked-up predicate which is true of both the inner and outer domain). So we need a 3rd domain which is outer the outer domain. And so on. You say you can stop at level 2, so to speak, and block the regress. And you appeal to Twardovsky here: in order to make a distinction, we need a super-ordinate concept. We have one for existent and subsistent; but not for subsistent and non-subsistent. Hum. So what is the super-ordinate for existent and subsistent? "Being an intentional object" I suppose? But if that is the case, how come non-items are not intentional objects? If it is by definition, then you presuppose what you want to show. If not, is it because they are unimaginable in some sense? In some sense, then, you have to say that non-items are un-namable (contrary to appearances). I guess you make the point with "the Nothing", but I got lost in the argument.
The alternative (that you do not discuss) is the way of Free Logic: you divorce the quantification domain from the naming domain. There are things you can name, that you will not quantify over. Of course, technically, it messes up things, because classical theory of quantification breaks down. And we need another one. But I think this is the promising road, as opposed to ruling out at the syntactical level some referring expressions. From the point of view of Free Logic, the quantification domain is what, by definition, contains all the existents (that is the philosophical starting point). There is leeway for interpreting this (you can be nominalist or more relaxed about who exists). But the crucial point is that you can name things outside the domain. And say true things about them. Including, of course, that they do not exist (i.e. are not part of the quantificational domain).

That leads me to a more profound/obscure reflection, concerning the dynamics of items/nonitems. In footnote 3, you virtually agree that some nonitems can become items, i.e. some names like Snazzo were not loyal, but have been tamed at some point, when Kripke's example became famous enough, I suppose, and so they are now loyal nonexistents. That is, I think, a very interesting phenomenon. But this dynamic is deeply anti-parmenidian. My understanding of Parmenides is not so much in the fact that he divided the ontological space into 2 (that which you relax to become a soft Parmenidian, allowing for a tripartition of the ontological space), but the fact that he says your either in or out. In the Sophist, that is what the parricide is all about: Plato/the Stranger treats the Sophist as someone who introduces some nonbeing into being, and Parmenides precisely said that nothing can change category, from a logical point of view. Then Plato/the Stranger adds some dynamic to his metaphysical picture, trying to accommodate Parmenides the static and Heraclites the dynamic. So if I am correct about this, then your picture is deeply anti-parmenidian, for it allows some nonitems to rise up to the standard of loyal nonexistent, under some specific conditions that you do not discuss in this paper. The label "soft paremenidian" is thus, to my ear, not accurate. It is rather in the wake of Plato's parricide.
Apart from the label issue. I think this dynamic phenomenon is very interesting. And I wonder how this can be empirically investigated: under what precise conditions the comparative existence judgements could change?

About the empirical results. Is "A Meinongian-Platonic anti-Parmenidean fan of the idea of ways of being" supposed to refer to Gabriel Markus?

What is wrong with (15)? I'd say it is indeed true. Did you test it and people say it's false? Cause in the examples above, you took political people who are not famously in a fiction (though I'm sure they are some many fictions).
By the way, according to Stacie, since fictions are about the real world, I think she would be happy with your results by interpreting "exist" as "inhabiting the real world".

I'm naturally going to push the anti-realist line at the end.
The metalinguistic story need not be intellectualist in the way you suggest. I think Donnellan's metalinguistic story about empty names is not difficult to ascribe to folks. When you come up with a name, and you are asked whether the referent of the name exist, you look for the origin of the name. Now, this historico-causal chain can either end in a block or not. Interestingly though (and going beyond Donnellan) we distinguish between several kinds of blocks. Some are fictional works: very distinctive kinds of blocks. Some are administrative mis-spellings. Very different from fiction (no pretence involved at all).
And finally, as a rejoinder for the anti-luxiriantist, "exist" in your experiment is contextual in the sense that it means "either the name refer or it pretends to refer" (which roughly means exist in reality or in fiction). And so you have the grouping of real and fictional names together, and exclude failed names. But that is compatible with the idea that fictional names do not refer, for they only pretend refer.

Again: hope this helps!
And thanks for the sharing of your paper,
Best,

Reviewer 2 Report

This is an interesting contribution to the literature on non-existence.  I am recommending for publication with a couple of minor revisions.  

1.

Lines 281-282 'she still has a higher degree of existence in terms of its abstractedness than Moloch 281 and Snazzo.'

- I find the expression 'higher degree of existence in terms of its abstractedness' to be a bit confusing.  Simply deleting 'in terms of its abstractedness' would eliminate the confusion.  Or finding another way to say what the author wants to say that doesn't carry the suggestion that existence comes in degrees of abstractedness and the difference between Emma Bovary and Moloch is that Emma's existence is more abstract than Moloch's existence. 

2.

Footnote 8  “Ron Wesley” 

- This is a typo, right?  I take it the author means 'Ron Weasley'.    
